# Tularemia Outbreaks in Spain from 2007 to 2020 in Humans and Domestic and Wild Animals

**DOI:** 10.3390/pathogens10070892

**Published:** 2021-07-14

**Authors:** Olga Mínguez-González, César-Bernardo Gutiérrez-Martín, María del Carmen Martínez-Nistal, María del Rosario Esquivel-García, José-Ignacio Gómez-Campillo, Jesús-Ángel Collazos-Martínez, Luis-Miguel Fernández-Calle, Cristina Ruiz-Sopeña, Sonia Tamames-Gómez, Sonia Martínez-Martínez, Constantino Caminero-Saldaña, Marta Hernández, David Rodríguez-Lázaro, Elías-Fernando Rodríguez-Ferri

**Affiliations:** 1Servicio de Sanidad Animal, Dirección General de Producción Agropecuaria e Infraestructuras Agrarias, Consejería de Agricultura, Ganadería y Desarrollo Rural, Junta de Castilla y León, 47007 Valladolid, Spain; MinGonOl@jcyl.es (O.M.-G.); ext-FerCalLu@jcyl.es (L.-M.F.-C.); 2Departmento de Sanidad Animal, Facultad de Veterinaria, Universidad de León, Campus de Vegazana s/n, 24007 León, Spain; smarm@unileon.es (S.M.-M.); ef.rferri@unileon.es (E.-F.R.-F.); 3Laboratorio Regional de Sanidad Animal, Consejería de Agricultura, Ganadería y Desarrollo Rural, Junta de Castilla y León, Villaquilambre, 24500 León, Spain; MARNISCA@jcyl.es (M.d.C.M.-N.); esqgarro@jcyl.es (M.d.R.E.-G.); gomcamjo@jcyl.es (J.-I.G.-C.); ext-ColMarJe@jcyl.es (J.-Á.C.-M.); 4Servicio de Vigilancia Epidemiológica y Enfermedades Transmisibles, Dirección General de Salud Pública, Consejería de Sanidad, Junta de Castilla y León, 47007 Valladolid, Spain; RuiSopCr@jcyl.es (C.R.-S.); TamGomSo@jcyl.es (S.T.-G.); 5Área de Plagas, Instituto Tecnológico Agrario de Castilla y León (ITACyL), 47009 Valladolid, Spain; CamSalCo@itacyl.es; 6Laboratorio de Biología Molecular y Microbiología, Instituto Tecnológico Agrario de Castilla y León (ITACyL), 47009 Valladolid, Spain; hernandez.marta@gmail.com; 7Unidad de Microbiología, Departamento de Biotecnología y Ciencia de los Alimentos, Facultad de Ciencias, and Research Centre for Emerging Pathogens and Global Health, Universidad de Burgos, 09001 Burgos, Spain; drlazaro@ubu.es

**Keywords:** *Francisella tularensis*, tularemia, lagomorphs, vole, shrew, tick, canids, human, sheep, dog, fox, wolf, crayfish, water

## Abstract

In this study, tularemia outbreaks associated with humans and several domestic and wild animals (Iberian hares, wild rabbits, voles, mice, grey shrews, sheep, dogs, foxes, wolves, ticks, and river crayfish) are reported in Spain from 2007 to 2020. Special attention was paid to the outbreaks in humans in 2007–2009 and 2014–2015, when the most important waves occurred. Moreover, positive rates of tularemia in lagomorphs were detected in 2007–2010, followed by negative results in 2011–2013, before again returning to positive rates in 2014 and in 2017 and in 2019–2020. Lagomorphs role in spreading *Francisella tularensis* in the epidemiological chain could not be discarded. *F. tularensis* is described for the first time infecting the shrew *Crocidura russula* worldwide, and it is also reported for the first time infecting wild rabbits (*Oryctolagus cuniculus*) in Spain. Serological positives higher than 0.4% were seen for sheep only from 2007–2009 and again in 2019, while serological rates greater than 1% were revealed in dogs in 2007–2008 and in wild canids in 2016. *F. tularensis* were detected in ticks in 2009, 2014–2015, 2017, and 2019. Lastly, negative results were achieved for river crayfish and also in environmental water samples from 2007 to 2020.

## 1. Introduction

*Francisella tularensis*, the etiological agent of the zoonosis tularemia, is a fastidious, aerobic gram-negative intracellular γ-proteobacterium with a small genome that is found in nature in association with a wide variety of animals, and it is considered a highly virulent risk 3 (biosafety level 3) pathogen [1,2,3]. Lagomorphs are the most common animal source for human infection, and ticks are the most important arthropod vectors although the number of species susceptible to infection by *F. tularensis* is higher than for any other known zoonotic organism [4]. Even so, large airborne and water-transmitted tularemia epidemics have also been reported [1,5]. *F. tularensis*, which has infectious doses as low as 10–50 CFU, comprises three subspecies: subsp. *tularensis* (or type A), subsp. *holartica* (or type B), and subsp. *mediasiatica*. Both type A and type B strains of *F. tularensis* can infect humans via direct contact with infected animals, ingestion of contaminated water or food, inhalation of contaminated aerosol, contact with contaminated soil or water environments, and arthropod bites (mainly ticks) [4,5,6]. *F. tularensis* subsp. *tularensis* can even cause life-threatening disease, and its distribution is mainly limited to North America although some studies have reported its appearance in Europe, where the first description occurred in Slovakia in 1998 [7,8]; however, further cases have not been reported. *F. tularensis* subsp. *holarctica* (or type B) produces a less severe disease, and it is linked to disease in rodents and hares [9]. This subspecies has widely spread throughout the Northern Hemisphere and has a genetic diversity more restricted than subsp. *tularensis* [4]. This latter feature seems to suggest a recent emergence as well as successful geographic spread [10,11,12]. In addition, *F. tularensis* subsp. *holarctica* was recently detected in Australia (mostly in Tasmania and Sydney) in the 2010s [13,14].

The infection caused by *F. tularensis* was first reported in Spain in late 1997, when a significant human outbreak related to hunting and handling of hares was diagnosed in Castile and León, northwest Spain [15,16]. The most common clinical form was ulceroglandular tularemia, followed by glandular and typhoid forms. A second human outbreak also took place in the same region between 2007 and 2008, after 10 years of no epidemiological activity, linked to common vole overcrowding, and the most prevalent clinical forms in this case were typhoidal and pneumonic tularemia [4,17,18]. The infection source was through inhalation. Furthermore, other cases associated with crayfish fishing were also described during the 2000s in Spain [19].

The aim of this study was to report the outbreaks of tularemia suffered in northwest Spain from 2007 to 2020, to determine its prevalence in humans and several domestic and wild animals, and to emphasize some epidemiological findings observed in Spain for the first time.

## 2. Materials and Methods

### 2.1. Sampling

The study was conducted in Castile and León, a Spanish region located in the northwest quadrant of the Iberian Peninsula. This region is divided into nine provinces: Ávila, Burgos, León, Palencia, Salamanca, Segovia, Soria, Valladolid, and Zamora (Figure 1). All samples were collected as part of a regional surveillance program of wildlife species for different zoonoses, and the sampling units were the Official Local Veterinary Units and the Agrarian Technological Institute of Castile and León (ITACyL). Selection criteria were revised each year, taking into account the occurrence of human cases of tularemia and density of common voles.

### 2.2. Isolation and Characterization of Francisella tularensis from Animals or Water

#### 2.2.1. Culture and Biochemical Tests

The *Francisella tularensis* subsp. *holarctica* isolates were recovered at the Laboratorio Regional de Sanidad Animal, Castile and León, dependent on the Consejería de Agricultura y Ganadería de la Junta de Castilla y León, according to the methods described by the Laboratorio Central de Veterinaria, Algete (Madrid, Spain) (dependent on the Spanish Ministry of Agriculture), such as isolation, identification, immunological, and molecular methods, based in turn on the World Health Organization [20]. Samples from hares, wild rabbits, voles, shrews, ticks, or crayfish were recovered from live animals or cadavers. The organs chosen were the liver and spleen, except for ticks, which were processed as a whole, and crayfishes, from which the cephalothorax was removed (Table 1). In addition, samples from water were also taken, but only PCR detection was carried out because *F. tularensis* culture from water samples remains extremely difficult. Biological security measures for biological agents of class 3 were used for isolation and culture. On the other hand, no authorization for handling *F. tularensis* subsp. *holarctica* is required in Spain because this subspecies is a group 2 biological agent.

Samples were cultured at 37 °C with 5% CO_2_ for 2 days on modified Thayer–Martin agar plates containing GC medium base (36 g/L), hemoglobin (10 g/L), and Vitox supplement (2 vials/L; Oxoid Ltd., Hampshire, England). After isolation of the colonies morphologically compatible with *F. tularensis* (mucous grayish colonies that coalesce quickly in strand shape), Gram staining (stained weakly, gram-negative small coccobacilli) and other biochemical tests, such as oxidase and catalase activities, glucose and glycerol fermentations, and urea hydrolysis, were carried out.

#### 2.2.2. Detection of *Francisella tularensis* by Real-Time PCR

For molecular characterization, *the Francisella* genus was identified by conventional and real-time PCR specific for the *fopA* gene [21]; for the recognition of *F. tularensis* subsp. *holartica*, PCR specific for the *tul*4 gene was carried out, as described previously [22].

#### 2.2.3. Detection of *Francisella tularensis* by Immunological Tests

For immunological characterization, a microagglutination test (MAT) was carried out using sera from mice, sheep, hunting dogs, foxes, and voles. Fifty microliters of test sera was 1:10 diluted in a U-bottom 96-microwell plate, and then two-fold dilutions until 1:640 were carried out. Afterwards, 25 μL of commercial inactivated antigen was dispensed in each well, and the microplate was shaken at 37 °C for 24 h before culture. An irregular agglutination in the walls of the well from a titer ≥80 was taken as a positive reaction (except for sheep, in which the samples were considered as positive from a titer ≥20), while a well-formed button of the antigen at the bed of the well was considered a negative reaction.

### 2.3. Isolation and Characterization of Francisella tularensis from Humans

Suspected cases in humans emanated from nonspecific hyperthermia and lymphadenopathies along with a clinical history of contact with animals. For culture diagnosis, the method described for animals or water was also applied to human samples. Liver, spleen, pus, and respiratory samples or gastric aspirates were typically taken (Table 1). In order to characterize the isolates, biochemical tests, and antigen detecting tests, PCR or other molecular methods already cited in the paragraph above were used.

Among the immunological methods, a MAT for the detection of IgG and IgM was again conducted, taking into consideration that serum antibodies do not reach detectable levels until 10 days, at least after the first symptoms of the disease. A seroconversion (fourfold the original titers) or titers <80 were considered suspicious, while titers ≥80 were deemed positive. The positivity criterion established by the Laboratorio Central de Veterinaria, Algete (Madrid, Spain), dependent on the Spanish Ministry of Agriculture, was followed for MAT.

As an alternative, a direct ELISA or a capture ELISA using monoclonal antibodies against the *F. tularensis* lipopolysaccharide was used [23,24].

For molecular methods, a PCR based in *tul*4 gene, which encodes an outer membrane protein, was used [25]. The considerations made on real-time PCR in the case of animals or water were equally applied to humans [26].

## 3. Results and Discussion

### 3.1. Outbreaks in Humans

A total of 507 cases (20.50 per 100,000 inhabitants) were confirmed in 2007 (Table 2); 91.5% by MAT; 5% by culture, isolation, and identification; and 3.5% by PCR, whereby 59.6% of them were grouped between June and August. The province where most cases were seen was Palencia (157.18 per 100,000 inhabitants), followed by Zamora (37.51 per 100,000 inhabitants). Notable differences were seen by gender (with a male predominance of almost 80.0%). The epidemiological survey revealed that most cases (31%) matched to outdoor workers in contact with gardens or natural environments, followed by those caused by contact with rodents (21%), domestic dogs or cats (17%), and crayfish (11%) as well as those who performed common trips to the countryside and were bitten by arthropods (11%). Rates less than 10% were linked to contact with livestock, manure, straw, or alfalfa as well as hare handling (Figure 2). The most common clinical form was typhoidal (71.6%, which is usually characterized by a severe disease with high fever and confusion, occurring through different modes of infection), followed by ulceroglandular (14.6%, characterized by skin ulcers with regional lymphadenopathy), glandular (12.6%, with only regional lymphadenopathy), and oculoglandular (1.2%, related to conjunctivitis and preauricular lymphadenopathy). In this context, Allue et al. [17] stated that harvesting tasks trigger aerosols being able to carry *F. tularensis*. Another hypothesis by this same group referred to environmental conditions (mild winters and dry springs) that could have contributed to tularemia outbreaks along with a reservoir and infection source diversity [17].

A total of 153 cases (6.12 per 100,000 inhabitants) were informed in 2008, with Palencia having the highest number of cases (24.36 per 100,000 inhabitants), followed by Zamora (13.18 per 100,000 inhabitants) and Soria (6.41 per 100,000 inhabitants) (Table 2). Major differences were again seen by gender (64.7% in men, among which 40–44 years was the age group most affected). The most current clinical form continued to be typhoidal (27.7%), followed by lymphoid (16.1%). Although the most common exposition factors were agriculture and gardening tasks (37.9%), the main factor linking outbreaks (34.8%) was handling and/or flaying of Iberian hares. In an epidemiological study conducted in 2008, cases were related to vole density; however, a coincidence in the geographical distribution of their intensities was not seen [17]. In addition, there was no evidence of striking differences in cases according to the acquisition route (respiratory tract or by contact), to the vole plague distribution, or to hydrography [17].

A relative epidemiological silence was observed between 2009 and 2013 in Castile and León because only 25 cases (1 per 100,000 inhabitants) were informed in 2009 in addition to four (0.16 per 100,000 inhabitants) in 2010, three (0.12 per 100,000 inhabitants) in 2011, and two each in 2012 and 2013 (0.08 per 100,000 inhabitants) (Table 2). The most common cases in these years were reported in men (up to 100% in 2011 and 2013) between 50 and 54 years old (20%). The clinical forms related to contact were more prevalent (50.0%) than those transmitted by a respiratory route (36.1%). The most common form of exposure was contact with cadavers (44.4%), mainly related to hare handling and/or flaying.

This declining trend was altered in August 2014, when 112 cases (4.48 per 100,000 inhabitants) were informed (mostly men, 72.3%), but a lesser number (1.24 per 100,000 inhabitants) was seen in 2015. In this latter year, the ulceroglandular form (29.0%) was the most common, and Iberian hare contact was again the most prevalent risk factor (38.7%). Lastly, three cases were notified (0.12 per 100,000 inhabitants) in 2016 in addition to 15 (0.60 per 100,00 inhabitants) in 2017 and eight (0.32 per 100,000 inhabitants) in 2018. We were not offered data from 2019 or 2020 (Table 2).

### 3.2. Outbreaks in Lagomorphs (Iberian Hares—Lepus granatensis—And Wild Rabbits—Oryctolagus cuniculus)

A total of 17.3% *Lepus granatensis* samples were positive (34 cases) for *Francisella tularensis* from the 197 sampled (mostly cadavers) between December 2006 and December 2007. The first two cases in *Oryctolagus cuniculus* were notified in 2008, when 21.3% (42/197) of lagomorphs were positive. In view of the results of human and hare cases, it could be stated that lagomorphs might have acted as reservoirs of human tularemia in the outbreak that occurred in 2007–2008. A marked decline in tularemia in lagomorphs was observed in 2009, a year in which positivity in animals decreased to 14.6% (7/48). Six of the nine provinces in Castile and León were sampled, giving rise to positive animals, especially Zamora, with 71.4% hares. The overall analyses indicated that 23.3% of hares (7/30) were positive in this year (Table 3).

This tendency halved in 2010 because only 7.4% (2/27) of lagomorphs were positive, and all results were negative in 2011–2013. However, 21.9% of Iberian hares (18/82) were positive for tularemia in 2014. The rate of positive cases in wild rabbits was 11.4% (5/44) in this same year, and this finding would justify the alert of epidemiological risk, at least for these lagomorphs.

The overall positivity percentage in 2015 was 8.1% (19/235), about 10 points lower than the previous year. The fall was amazing in 2015, and global tularemia rates (8.1%) only remained in the Palencia province. In view of the results obtained in 2014 and 2015, it could be speculated that *F. tularensis* could have evolved to also accommodate alternative hosts, such as wild rabbit, but certainly without losing the relevant role played by hares. Most cases in wild rabbits were detected in 2015 in cold months (January and February), while those detected in hares were also seen in April and June. On the other hand, it is notable that tularemia is reported here for the first time in *Oryctolagus cuniculus* in Spain.

Only four positive cases (1.2%, 4/317) were seen in lagomorphs in 2016 (three from wild rabbits and one from Iberian hares) along with another four in 2017 (0.04%, 4/1078), all belonging to *Lepus granatensis*. No positive cases were seen from any of the lagomorphs tested in 2018 (Table 3).

Palencia showed by far the highest tularemia prevalence in lagomorphs (over 20%) between 2007 and 2015. Other provinces, such as Valladolid and Salamanca, also provided important results (~15%). From almost 700 hare samples collected in the last decade in Castile and León, a mean positivity of 18.0% was reached, with a peak value of 28.2% in 2008. On the other hand, the role of wild rabbits must not be discarded as an epidemiological amplifier, having achieved values of 7.7% in 2005 and up to 11.4% in 2014, probably matching with its major abundance in these favorable years.

Lastly, low percentages were reported in lagomorphs in 2019 and 2020, with 3.8% (14 cases from 367 samples: 13 from Iberian hares and 1 from wild rabbits) in 2019 and only 2.3% (seven cases from 301 samples, all coming from *Lepus granatensis*) in 2020 (Table 3).

### 3.3. Outbreaks in Common Voles (Microtus arvalis)

A 1.9% positivity in *Microtus arvalis* (12/646) was recorded in 2007. Over a total of 339 samples collected in 2008, no positive cases were detected. Despite the lack of cases observed in 2008 compared to 2007, surveillance was kept during the subsequent years. Thirty-eight common voles were collected in 2009, but none were positive; however, from the samples gathered in 2010, the positivity was 14.7% (5 from 34 samples) (Table 3).

The sample size increased in 2011, and surveillance pressure was kept, especially in Palencia, Segovia, Valladolid, and Zamora, which were considered as at-risk provinces. A total of 207 *Microtus arvalis* samples were collected in these four provinces, all of which were negative. Only 72 voles were caught in 2012, all of which were negative. In the absence of positive cases, the sampling focused mainly on at-risk provinces, revealing once more that Palencia was subjected to the highest number of analyses in 2013, constituting 220 of the 292 samples (75.3%) in Castile and León, all of which were negative.

After three years without cases of tularemia, the attention was refocused in 2014, when 11.6% of isolations (160/1,439) were identified as positive in common voles in Castile and León (mostly recovered from Palencia), similar to the rates recorded in 2010 but far above those obtained in 2007, i.e., the previous outbreak [27]. Only 92 samples were collected in 2015 of which one was positive by both isolation culture and PCR; therefore, minimal positivity rates (1.1%) were observed. The results found for *Microtus arvalis* in 2015 were lower than those obtained for lagomorphs. Percentages less than 2.3% were seen between 2016 and 2020 except in 2018, during which the results were negative. In an overall study from 2007 to 2016, 3.1% positivity was reached, with Soria having the highest rate (6.7%). Ten cases from 751 samples (1.3%) were reported again in 2019 in Palencia along with only two from 138 samples (1.4%) in 2020 in Salamanca. In short, while the difficult years in *Microtus arvalis* were 2010 and 2014, the critical years in *Lepus granatensis* were 2007–2009 and 2014 (Table 3).

The finding that no cases were detected in voles in 2008, and a scarce number was obtained in 2015, while several outbreaks of tularemia were observed to be strongly related to humans in these same years, suggests that reservoirs other than common voles (e.g., lagomorphs) contribute to human tularemia in contrast to the hypotheses defended by other authors in Spain [28], who postulated that *Microtus arvalis* acts as a key spillover host of *F. tularensis* in northwest Spain.

### 3.4. Outbreaks in Other Micromammals (Mice,—Mainly Apodemus Sylvaticus and Algerian Mice—Mus Spretus, and Gray Shrews—Crocidura Russula)

Thirteen specimens from other micromammals (OMMs) were collected in 2009 in which field mice and shrews were included, but none were positive; this result was also observed between 2010 and 2013. After five years without *Francisella tularensis* isolations, 6 of 45 samples (13.3%) tested positive in 2014. These results confirmed the presence of a new health emergency in animals, supported by the values of the risk species (rodents, Soricidae, and lagomorphs). In addition, they evidence the evolution in disease epidemiology, now encompassing other actors, such as hares, common voles, field rabbit, other small field rodents, and shrews. This reveals a shy but decisive adaptation of *F. tularensis* to new hosts in these areas, although hares remained the main vector reservoirs.

Only 34 samples (17 from field mice and 17 from shrews) were collected in 2015, and only one mouse was positive by PCR (5.9% of the mouse samples tested) (Table 3). Surprisingly, five positive isolations were recovered in Palencia in 2016 from a total of 560 samples (0.9%), four of them from shrews and one from a field mouse. At this point, it must be highlighted that the two cases reported from *Crocidura russula* in 2014 and the six described from this species in 2016 are the first cases observed worldwide as reservoir hosts of *F. tularensis*. No positive cases from OMMs were detected between 2017 and 2018. Lastly, only 11 cases (1.9%, four from mice and seven from shrews) were registered in Palencia in 2019 from a total of 580 samples recovered in Castile and León, while no cases were reported in OMMs from a total of 271 specimens taken in 2020 (Table 3).

### 3.5. Surveillance in Other Animal Species of Minor Importance

#### 3.5.1. Sheep

As part of the tularemia surveillance plan, serological assays in sheep began in 2007 when the study, coincident with risk area herds, reached 64,904 samples from 1257 livestock farms in León (20.4%), Palencia (40.7%), Valladolid (20.8%), and Zamora (18.1%). These samples were collected for the monitoring of diseases of obligatory declaration in Castile and León. The overall seroprevalence was of only 0.4%, and a drop in the number of positive animals was seen in 2009, reaching only 0.1% (Table 3).

This same trend was observed in 2010 and 2011, while no positive cases were detected from 2012 to 2016. Only 0.04% of sheep samples (4/9508) were positive in 2017, and all sera were again negative in 2018. The percentage of positive sera in 2019 (0.9%, 111/12,560) and 2020 (0.03%, 1/3958) was again very limited (Table 3). These data stated the existence of an epidemiological indicator that could be useful for surveillance purposes. However, it must be noted that there were no data of a second serological test in the same animal, and consequently, no seroconversion could be observed.

#### 3.5.2. Dogs

A total of 528 dogs belonging to risk groups were tested in 2007 by MAT, and 5.1% were positive, being good indicators of *Francisella tularensis* presence. In this way, seroprevalences of up to 37.0% in dogs belonging to specific risk groups in an endemic area of tularemia in Slovakia, with the peculiarity of its high persistence, were reported [29], and dogs are considered good indicators of active natural outbreaks and suitable markers for its surveillance. A mean of 1018 dogs were monitored in our region in 2008–2009 and 2014–2016, with very low positivity rates (1.0%, 5/472, in 2008 and 0.1%, 1/1232, in 2015), while no positive results were found between 2010 and 2013, in 2017, and in 2019 from a mean of 961 samples. Lastly, only one dog from Palencia was found positive in each of the years 2018 and 2020 (0.0008% and 0.12%, respectively) (Table 3). This context confirms its role as an indicator or disease sentinel.

#### 3.5.3. Foxes (*Vulpes vulpes*) and Wolves (*Canis lupus signatus*)

A scarce importance of foxes and wolves was shown in our environment. Although almost 600 animals were sampled in Castile and León between 2011 and 2016, positive sera were found in one wolf (0.01%) in 2015 and in two (also 0.01%) the next year. In addition, a total of 802 foxes or wolves were tested from 2017 to 2020, but no positivity was encountered (Table 3). This low significance only reflects an indirect form (by means of antibody presence) of *Francisella tularensis* traffic in the periods of greatest epidemic risk in other species, surely inspired by their predator state or occasional contact with hares, wild rabbits, or rodents.

#### 3.5.4. Invertebrate Vectors

Ticks recovered from specifically sampled live hosts (primarily parasitizing voles, ovine, or hares) were tested microbiologically. Sampling was started in 47 ticks in 2007, while 30 samples were tested in 2008, all of which were negative. However, 8.3% (3/36) of ticks were recorded as positive in 2009 (Table 3). Four of the 11 ticks (36.4%) tested in 2014 were PCR-positive, and seven were positive from the 10 samples (70%) collected 1 year later. It must be highlighted that all positive cases in 2014 and 2015 were in ticks that were parasitizing the sampled lagomorphs. In this way, one of the first isolates coming from a vole in the first outbreak (1997–1998) was accompanied by tick isolation, and from these, *Francisella tularensis* subsp. *holarctica* was also recovered, as seen with other ticks sampled during the same year [8]. No ticks were sampled for the next 11 years; however, *F. tularensis* subsp. *holartica* was isolated from two and four ticks investigated in 2017 and 2019, respectively (Table 3).

#### 3.5.5. River Crayfish (*Procambarae alleni*)

Following the outbreak that took place in Castilla-La Mancha, Spain [18], along with some cases seen in Palencia, sample collection and further study of river crayfish were scheduled, conducting the determination by culture or PCR. The sampling sequence had special significance during 2007, when 256 samples were collected, all of which were negative. Only 12 specimens were collected in 2009 in addition to four in 2011 and five in 2014, all of which were negative (Table 3). These shellfish were caught in water streams belonging to at-risk areas.

### 3.6. Francisella tularensis in Environmental Water Samples

Fifty-nine samples were collected in 2007, all of which were negative. They were taken from different locations (wells, ponds, streams, canals, etc) in at-risk areas, in coincidence with positive or suspicious cases in wildlife animals. All results were also negative for the 18 and 11 specimens taken in 2008 and 2009. No samples were collected from 2010 to now (Table 3).

## 4. Conclusions

This study supported the outbreaks of tularemia in humans, lagomorphs, and common voles suffered in Castile and León, northwest Spain, from 2007 to 2020. The reemergence of this disease after confirming the presence of *Francisella tularensis* in hares, wild rabbits, common voles, and to a lesser extent, field mice, shrews, and ticks and the results of the epidemiological surveillance with low titers of antibodies to *F. tularensis* in sheep, dogs, and wild canids suggest a persistence of this pathogen in carriers because, except for the outbreak that occurred in Castilla-La Mancha (Spain) in crayfish, no other episode has taken place outside of Castile and León. Before now, the animal species or environmental niche representing the real reservoir of *F. tularensis* could not be ascertained. Lastly, two highlighted findings are reported here for the first time: the presence of tularemia in wild rabbits in Spain and that in the shrew *Crocidura russula* worldwide.

## Figures and Tables

**Figure 1 pathogens-10-00892-f001:**
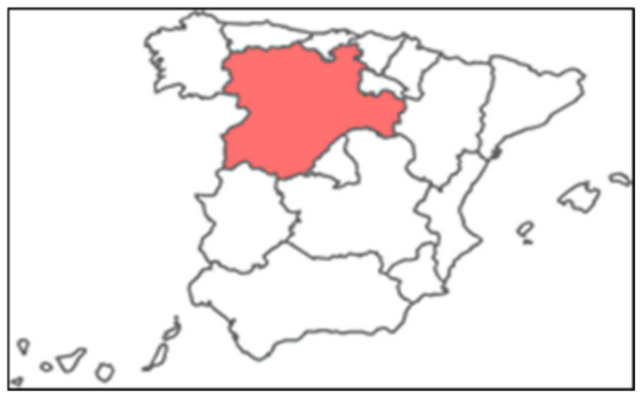
Castile and León region (northwest Spain).

**Figure 2 pathogens-10-00892-f002:**
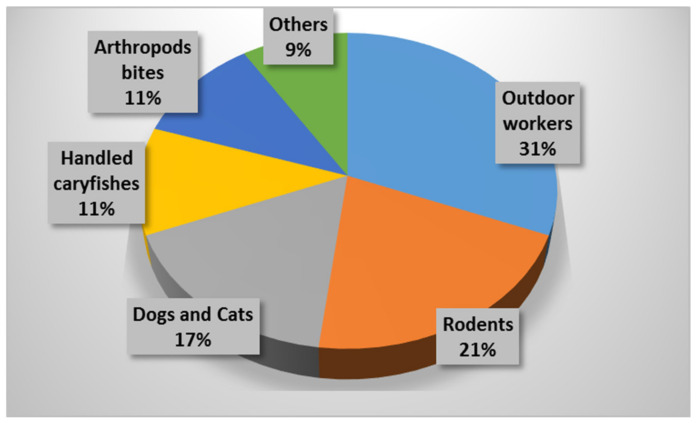
Epidemiological survey of human tularemia reported in Castile and León (Spain) in 2007.

**Table 1 pathogens-10-00892-t001:** Sample type and specific test(s) performed on human, domestic and wild animals, and water samples in this investigation.

Source	Organs/Tissues	Type(s) of Methods Performed
Iberian hares	Liver and spleen	Culture, PCR *
Wild rabbits	Liver and spleen	Culture, PCR
Voles	Liver and spleen	Culture, PCR
Mice	Liver and spleen	Culture, PCR
Algerian mice	Liver and spleen	Culture, PCR
Gray shrews	Liver and spleen	Culture, PCR
Sheep	Sera	MAT **, ELISA ***
Dogs	Sera	MAT, ELISA
Foxes	Sera	MAT, ELISA
Wolves	Sera	MAT, ELISA
River crayfish	Cephalothorax	MAT, ELISA
Ticks	Whole	MAT, ELISA
Water	---	MAT, ELISA
Humans	Liver, spleen, pus, respiratory samples, gastric aspirates, sera	Culture, PCR, MAT, ELISA


* polymerase chain reaction; ** microagglutination test; *** enzyme-linked immunosorbent assay.

**Table 2 pathogens-10-00892-t002:** Number of samples positive with *Francisella tularensis* subsp. *holartica* from the specimens sampled in Castile and León from human sources between 2007 and 2020.

Province or Gender	Year of Study Number of Cases (Rate per 100,000 Inhabitants)													
	2007	2008	2009	2010	2011	2012	2013	2014	2015	2016	2017	2018	2019	2020
Castile and León	507 (20.50)	153 (6.12)	25 (1.00)	4 (0.16)	3 (0.12)	2 (0.08)	2 (0.08)	112 (4.48)	31 (1.24)	3 (0.12)	15 (0.60)	8 (0.32)	ND *	ND
Ávila	2 (1.20)	2 (1.20)	0 (0)	0 (0)	0 (0)	0 (0)	0 (0)	0 (0)	0 (0)	0 (0)	0 (0)	1 (0.60)	ND	ND
Burgos	30 (8.70)	17 (4.93)	2 (0.58)	0 (0)	1 (0.29)	1 (0.29)	0 (0)	6 (1.74)	3 (0.87)	0 (0)	0 (0)	0 (0)	ND	ND
León	49 (9.8)	18 (3.6)	2 (0.40)	0 (0)	1 (0.20)	0 (0)	0 (0)	6 (1.20)	1 (0.20)	1 (0.20)	1 (0.20)	0 (0)	ND	ND
Palencia	271 (157.18)	42 (24.36)	3 (1.74)	1 (0.58)	0 (0)	0 (0)	1 (0.58)	84 (48.72)	9 (5.22)	0 (0)	5 (2.90)	3 (1.74)	ND	ND
Salamanca	6 (1.68)	10 (2.80)	2 (0.56)	1 (0.28)	0 (0)	1 (0.28)	0 (0)	1 (0.28)	2 (0.56)	0 (0)	0 (0)	1 (0.28)	ND	ND
Segovia	1 (0.62)	1 (0.62)	2 (1.24)	0 (0)	0 (0)	0 (0)	0 (0)	0 (0)	1 (0.62)	0 (0)	0 (0)	0 (0)	ND	ND
Soria	6 (6.41)	6 (6.41)	2 (2.14)	1 (1.07)	0 (0)	0 (0)	0 (0)	0 (0)	3 (3.20)	1 (1.07)	0 (0)	1 (1.07)	ND	ND
Valladolid	68 (13.46)	31 (6.14)	10 (1.98)	0 (0)	1 (0.19))	0 (0)	1 (0.19)	8 (1.58)	6 (13.46)	0 (0)	5 (0.99)	0 (0)	ND	ND
Zamora	74 (37.51)	26 (13.18)	2 (1.01)	1 (0.51)	0 (0)	0 (0)	0 (0)	7 (3.55)	6 (3.04)	1 (0.51)	4 (2.03)	2 (1.01)	ND	ND
Female	102 (20.1%)	54 (35.3%)	6 (24.0%)	1 (25.0%)	0 (0%)	1 (50.0%)	0 (0%)	31 (27.7%)	14 (45.2%)	0 (0%)	2 (13.3%)	5 (0.62%)	ND	ND
Male	405 (79.9%)	99 (64.7%)	19 (76.0%)	3 (75.0%)	3 (100%)	1 (50.0%)	2 (100%)	81 (72.3%)	17 (54.8%)	3 (100%)	13 (86.7%)	3 (0.38%)	ND	ND

* No available data.

**Table 3 pathogens-10-00892-t003:** Number of samples positive with *Francisella tularensis* subsp. *holartica* from the specimens sampled in Castile and León from different animal sources between 2007 and 2020.

Source	Year of Study (Positive Samples/Number of Samples) (Percentage)													
	2007	2008	2009	2010	2011	2012	2013	2014	2015	2016	2017	2018	2019	2020
Lagomorphs	34/216 (15.7%)	42/197 (21.3%)	7/48 (14.6%)	2/27 (7.4%)	0/222 (0%)	0/84 (0%)	0/115 (0%)	23/126 (18.2%)	19/235 (8.1%)	4/317 (1.2%)	4/1078 (0.04%)	0/302 (0%)	14/367 (3.8%)	7/301 (2.3%)
Iberian hares	34/197 (17.3%)	40/155 (25.8%)	7/30 (23.3%)	2/17 (11.8%)	0/35 (0%)	0/11 (0%)	0/60 (0%)	18/82 (21.9%)	9/78 (11.5%)	1/80 (1.2%)	4/82 (4.9%)	0/110 (0%)	13/272 (4.8%)	7/194 (3.6%)
Wild rabbits	0/19 (0%)	2/42 (4.8%)	0/18 (0%)	0/10 (0%)	0/187 (0%)	0/73 (0%)	0/55 (0%)	5/44 (11.4%)	10/157 (6.4%)	3/237 (1.3%)	0/256 (0%)	0/192 (0%)	1/95 (1.0%)	0/107 (0%)
Voles	12/646 (1.9%)	0/339 (0%)	0/38 (0%)	5/34 (14.7%)	0/207 (0%)	0/72 (0%)	0/292 (0%)	160/1439 (11.6%)	1/92 (1.1%)	23/1040 (2.2%)	3/545 (0.5%)	0/51 (0%)	10/751 (1.3%)	2/138 (1.4%)
Mice	---	---	0/11 (0%)	0/3 (0%)	0/115 (0%)	0/16 (0%)	0/69 (0%)	4/29 (13.8%)	1/17 (5.9%)	3/275 (1.1%)	0/102 (0%)	0/311 (0%)	4/359 (1.1%)	0/138 (0%)
Grey shrews	---	---	0/2 (0%)	0/4 (0%)	0/17 (0%)	0/12 (0%)	0/27 (0%)	2/16 (12.5%)	0/17 (0%)	6/422 (1.4%)	0/82 (0%)	0/132 (0%)	7/221 (3.2%)	0/133 (0%)
Sheep	281/64,904 (0.4%)	178/28,234 (0.6%)	27/27,527 (0.1%)	8/26,575 (0.03%)	4/30,522 (0.01%)	0/26,989 (0%)	0/2495 (0%)	0/2051 (0%)	0/2216 (0%)	0/2035 (0%)	4/9508 (0.04%)	0/7665 (0%)	111/12560 (0.9%)	1/3958 (0.03%)
Dogs	27/528 (5.1%)	5/472 (1.0%)	2/495 (0.4%)	0/559 (0%)	0/960 (0%)	0/939 (0%)	0/1131 (0%)	6/1361 (0.4%)	1/1232 (0.1%)	3/1528 (0.2%)	0/1304 (0%)	1/1207 (0.0008%)	0/875 (0%)	1/827 (0.12%)
Foxes and wolves	----	----	----	----	0/116 (0%)	0/53 (0%)	0/61 (0%)	0/71 (0%)	1/101 (0.01%)	2/190 (0.01%)	0/194 (0%)	0/197 (0.0%)	0/176 (0%)	0/233 (0%)
Ticks	0/47 (0%)	0/30 (0%)	3/36 (8.3%)	---	---	---	---	4/11 (36.4%)	7/10 (70.0%)	---	2/2 (100%)	---	4/4 (100%)	---
Crayfishes	0/256 (0%)	---	0/12 (0%)	---	0/4 (0%)	---	---	0/5 (0%)	---	---	---	---	---	---
Water	0/59 (0%)	0/18 (0%)	0/11 (0%)	---	---	---	---	---	---	---	---	---	---	---

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
