# Peer review of "Tularemia Outbreaks in Spain from 2007 to 2020 in Humans and Domestic and Wild Animals"

_pathogens, 2021, doi:10.3390/pathogens10070892_

Round 1

Reviewer 1 Report

This is an interesting study of human and animal tularemia in Castilla y Léon, Spain, since 2007.

Comments.

Page 2, line 49. F. tularensis subsp. novicida is currently considered a different species, F. novicida

Page 2, lines 51-52. "spreads by inhalation, direct contact with infected animals or ingestion of contaminated food or 52 water ". This sentence is unclear and should be split. Both type A and type B strains of F. tularensis can infect humans via direct contact with infected animals, ingestion contaminated water or food, inhalation of contaminated aerosols, contact with contaminated soil or water environments, and arthropod bites (mainly ticks).

Page 2, line 58. F. tularensis susp. holarctica has recently been detected in Australia (mainly Tasmania and Sidney).

Page 2, line 86: The authors should specify if any detention authorization of F. tularensis strains is required in Spain and give the corresponding authorization number/reference.

Page 3, lines 106-107. The authors must specify which Francisella species/subspecies can be amplified by the fopA gene, region of difference 1 (RD), and tul4 gene real-time PCR. Some of them are not specific for F. tularensis (e.g., the tul4 gene)

Page 4, line 142. "The most common clinical form was typhoid (71.6%) -suggesting that the main infection route was in-142 halation". This sentence is unclear. The typhoidal form of tularemia is usually characterized by a severe disease with high fever and confusion. It may occur through different modes of infection. However, the inhalation route of infection usually results in the pneumonic form rather than the typhoidal form of tularemia.

Page 4, lines 155-156. The authors should specify which criteria they used to define the "typhoidal forms" of tularemia and specify the clinical forms corresponding to the "lymphoid form."

Page 8, line 295: "Eslovaquia" should be "Slovakia"

Page 9, line 330. The authors should specify which methods were used to detect F. tularensis in water samples. They could mention that culturing F. tularensis remains extremely difficult even when PCR detects F. tularensis DNA.

A table summarizing culture, PCR, and serology results for the different types of tested animals/samples would be helpful. It isn't accessible from the text to understand which tests were performed (especially culture and PCR) for the different types of samples

English editing is needed

Author Response

REVIEWER 1

This is an interesting study of human and animal tularemia in Castilla y Léon, Spain, since 2007.

Comments.

Page 2, line 49. F. tularensis subsp. novicida is currently considered a different species, F. novicida

This information is already corrected in the revised manuscript: line 50 and following.

Page 2, lines 51-52. "spreads by inhalation, direct contact with infected animals or ingestion of contaminated food or 52 water ". This sentence is unclear and should be split. Both type A and type B strains of F. tularensis can infect humans via direct contact with infected animals, ingestion contaminated water or food, inhalation of contaminated aerosols, contact with contaminated soil or water environments, and arthropod bites (mainly ticks).

This sentence has been rewritten according to the suggestion by this reviewer: line 52 and following.

Page 2, line 58. F. tularensis susp. holarctica has recently been detected in Australia (mainly Tasmania and Sidney).

Two references have been added in the revised versión of the manuscript: lines 101 to 103.

Page 2, line 86: The authors should specify if any detention authorization of F. tularensis strains is required in Spain and give the corresponding authorization number/reference.

This requeriment is answered in the end of the manuscript, after “funding and acknowlegments”. Lines 386 to 389.

Page 3, lines 106-107. The authors must specify which Francisella species/subspecies can be amplified by the fopA gene, region of difference 1 (RD), and tul4 gene real-time PCR. Some of them are not specific for F. tularensis (e.g., the tul4 gene)

We agree with this reviewer. This requested information now appears in lines 114 to 116.

Page 4, line 142. "The most common clinical form was typhoid (71.6%) -suggesting that the main infection route was in-142 halation". This sentence is unclear. The typhoidal form of tularemia is usually characterized by a severe disease with high fever and confusion. It may occur through different modes of infection. However, the inhalation route of infection usually results in the pneumonic form rather than the typhoidal form of tularemia.

This suggestion has been taken into considerations and the rewritten sentence appears en lines 157 to 159.

Page 4, lines 155-156. The authors should specify which criteria they used to define the "typhoidal forms" of tularemia and specify the clinical forms corresponding to the "lymphoid form."

These forms are specified in the revised manuscript: lines 157 to 162.

Page 8, line 295: "Eslovaquia" should be "Slovakia"

Slovakia has been written in the revised manuscript: line 318.

Page 9, line 330. The authors should specify which methods were used to detect F. tularensis in water samples. They could mention that culturing F. tularensis remains extremely difficult even when PCR detects F. tularensis DNA.

This suggestion has been added in the revised manuscript: lines 98 to 100.

A table summarizing culture, PCR, and serology results for the different types of tested animals/samples would be helpful. It isn't accessible from the text to understand which tests were performed (especially culture and PCR) for the different types of samples.

This table (Table 1) appears now in the revised manuscript: lines 142 to 144.

English editing is needed

English has been now editing by MDPI Editorial Office.

Reviewer 2 Report

Interesting article describing the tularemia case trends in Spain since 2007. 

One main suggestion is to convert most graphs into one concise table. This would make it much easier for the reader to interpret the results as most of the graphs were difficult to see. Attached is a proposed layout for said table. 

Also, whenever you report a percentage in text, please also report the numerator and denominator (X/Y). 

Below are my specific line by line comments:

Line 43: lowercase the "G" in gram-negative

Line 46: Add a comma after vectors

Lines 61-62: Replace "one human" with "a" 

Line 76: Delete "Castilla y Leon" and the "( )" around CyL as this has already been abbreviated

Line 102: Lowercase the "G"s in gram and add a hyphen between gram and negative for consistency. Or delete the hyphen in the previous use

Line 135: 507 cases? Based on Figure 2, it appears that there were 470 cases at most. This is one of the reasons I suggested putting this data in a table to clear up some confusion. 

Line 137: Delete "of"

Lines 152-153: Reword to "... in 2008 (Fig. 2), with Palencia having the highest number of cases, followed by Zamora."

Line 153: If reporting a "significant difference", please also report the statistics and associated statistical output

Lines 157-158: Reword as "...outbreaks (34.8%) was handling and/or flaying of hares."

Line 158: Replace "other" with "an" and add a comma after 2008

Line 166: Delete "each of" 

Line 175: When reporting data for "years", please be consistent and report data for all years. It becomes confusing when your graphs only go up to 2018 and yet you discuss up through 2020 elsewhere in the manuscript. Another reason to convert results to a table 

Lines 183-184: This sentence is confusing as written. Please reword

Lines 193-196: Again, these sentences are confusing. Please reword

Lines 199-201: Again, this sentence is confusing. Please reword

Line 203: Change "notified" to "detected"

Line 204: Viral? Please specify

Lines 2019-221: Again, why is this data not in the graph in Figure 5? 

Line 223: Please reword to "In 2007, 1.9% of Microtus arvalis were reported as positive."

Line 226: Replace "was" with "were" 

Line 228: Thickness? Replace with, perhaps, "sample size"

Line 231: Reword to "... 2012, all of which were negative."

Line 234: Delete "Anyways, all resulted negative once more." Add "... in all CyL, of which all were negative."

Lines 240-241: ",... always below from those referenced to lagomorphs (Fig. 5)." is confusing. Please reword.

Lines 242-243: Please reword to "was reached, with Soria having the highest rate (6.7%)."

Line 251: Agent? Host would be more appropriate. F. tularensis is the agent and Microtus arvalis is the host of said agent. 

Line 253: Add "%" after the 0.85

Lines 254-255: Delete the hyphens after "sylvaticus, spretus, and russula. Add a space after the remaining hyphens 

Line 257: Replace "was" with "were"

Line 271: Denounced?

Line 284: Cancelled out? Please reword

Lines 286-287: Please reword to "The percentage of positive sera in 2019 (0.9%) and 2020 (0.03%) was very limited."  And again, why isn't this data reported in Figure 6?

Line 299: Canceled out? Again, please replace 

Lines 303 and 311: Italicize scientific names

Line 317: Reword to "in 2014, of which were all negative."

Line 322: Replace "studies were carried out" with "were sampled"

Line 323: Reword to "Three of the five ticks (60.0%) tested in 2014 were PCR positive..."

Line 331: Delete "..." and either replace with all categories or with ", etc."

Author Response

REVIEWER 2

Interesting article describing the tularemia case trends in Spain since 2007. 

One main suggestion is to convert most graphs into one concise table. This would make it much easier for the reader to interpret the results as most of the graphs were difficult to see. Attached is a proposed layout for said table. 

We agree with reviewer 2 and we have placed data not in one table, but in two tables: one for the human data and the other for the animal data. Human data are shown in landscape Table 2, as number of positive cases followeb by case rate per 100,000 inhabitans in the different provinces in Castilla y León, and also globally in Castilla y León (lines 198 to 199). Table 3 also appears in a landscape shape and includes animal data as number of positive cases / number of samples tested, followed by percentage for the different animals sampled (lines 202 to 203).

Also, whenever you report a percentage in text, please also report the numerator and denominator (X/Y). 

X/Y has been reported whenever possible.

Below are my specific line by line comments:

Line 43: lowercase the "G" in gram-negative

It has been corrected in the revised manuscript.

Line 46: Add a comma after vectors

It has been corrected in the revised manuscript.

Lines 61-62: Replace "one human" with "a" 

It has been corrected in the revised manuscript.

Line 76: Delete "Castilla y Leon" and the "( )" around CyL as this has already been abbreviated

It has been corrected in the revised manuscript.

Line 102: Lowercase the "G"s in gram and add a hyphen between gram and negative for consistency. Or delete the hyphen in the previous use

It has been corrected in the revised manuscript.

Line 135: 507 cases? Based on Figure 2, it appears that there were 470 cases at most. This is one of the reasons I suggested putting this data in a table to clear up some confusion. 

The number of cases (507) is certainly right. The problems is that the number of cases according to the dates was not clear enough. All has been solved in Table 2.

Line 137: Delete "of"

It has been corrected in the revised manuscript.

Lines 152-153: Reword to "... in 2008 (Fig. 2), with Palencia having the highest number of cases, followed by Zamora."

It has been rewritten in the revised manuscript.

Line 153: If reporting a "significant difference", please also report the statistics and associated statistical output

It has been corrected in the revised manuscript.

Lines 157-158: Reword as "...outbreaks (34.8%) was handling and/or flaying of hares."

It has been rewritten in the revised manuscript.

Line 158: Replace "other" with "an" and add a comma after 2008

It has been corrected in the revised manuscript.

Line 166: Delete "each of" 

It has been corrected in the revised manuscript.

Line 175: When reporting data for "years", please be consistent and report data for all years. It becomes confusing when your graphs only go up to 2018 and yet you discuss up through 2020 elsewhere in the manuscript. Another reason to convert results to a table 

It has been corrected in the revised manuscript through Tables 2 and 3

Lines 183-184: This sentence is confusing as written. Please reword

This sentence has been rewritten in the revised manuscript.

Lines 193-196: Again, these sentences are confusing. Please reword

These sentences have been rewritten in the revised manuscript.

Lines 199-201: Again, this sentence is confusing. Please reword

This sentence has been rewritten in the revised manuscript.

Line 203: Change "notified" to "detected"

It has been corrected in the revised manuscript.

Line 204: Viral? Please specify

It has been deleted in the revised manuscript.

Lines 2019-221: Again, why is this data not in the graph in Figure 5? 

It has been corrected in the revised manuscript through Table 3

Line 223: Please reword to "In 2007, 1.9% of Microtus arvalis were reported as positive."

It has been rewritten in the revised manuscript.

Line 226: Replace "was" with "were" 

It has been rewritten in the revised manuscript.

Line 228: Thickness? Replace with, perhaps, "sample size"

It has been rewritten in the revised manuscript.

Line 231: Reword to "... 2012, all of which were negative."

It has been rewritten in the revised manuscript.

It has been rewritten in the revised manuscript.

Line 234: Delete "Anyways, all resulted negative once more." Add "... in all CyL, of which all were negative."

It has been rewritten in the revised manuscript.

Lines 240-241: ",... always below from those referenced to lagomorphs (Fig. 5)." is confusing. Please reword.

This sentence has been rewritten in the revised manuscript.

Lines 242-243: Please reword to "was reached, with Soria having the highest rate (6.7%)."

It has been rewritten in the revised manuscript.

Line 251: Agent? Host would be more appropriate. F. tularensis is the agent and Microtus arvalis is the host of said agent. 

It has been rewritten in the revised manuscript.

Line 253: Add "%" after the 0.85

It has been corrected in the revised manuscript.

Lines 254-255: Delete the hyphens after "sylvaticus, spretus, and russula. Add a space after the remaining hyphens 

It has been corrected in the revised manuscript.

Line 257: Replace "was" with "were"

It has been corrected in the revised manuscript.

Line 271: Denounced?

It has been corrected in the revised manuscript.

Line 284: Cancelled out? Please reword

It has been corrected in the revised manuscript.

Lines 286-287: Please reword to "The percentage of positive sera in 2019 (0.9%) and 2020 (0.03%) was very limited."  And again, why isn't this data reported in Figure 6?

It has been rewritten in the revised manuscript.

Line 299: Canceled out? Again, please replace 

It has been corrected in the revised manuscript.

Lines 303 and 311: Italicize scientific names

It has been corrected in the revised manuscript.

Line 317: Reword to "in 2014, of which were all negative."

It has been corrected in the revised manuscript.

Line 322: Replace "studies were carried out" with "were sampled"

It has been corrected in the revised manuscript.

Line 323: Reword to "Three of the five ticks (60.0%) tested in 2014 were PCR positive..."

It has been rewritten in the revised manuscript.

Line 331: Delete "..." and either replace with all categories or with ", etc."

It has been deleted in the revised manuscript.

Round 2

Reviewer 1 Report

The manuscript has been much improved and is ready for publication.
As minor comments, the classic names for clinical forms of tularemia are: glandular, ulceroglandular, oculoglandular, oropharyngeal, pneumonic, and typhoidal.

Author Response

REVIEWER 1 (second revisión)

Thank you for your great opinion about our second version of manuscript “Tularemia Outbreaks in Spain from 2007 to 2020 in Humans and Domestic and Wild Animals”. The clinical names for clinical forms of tularemia have been modified in the second version of our manuscript.

Best regards.

César B. Gutiérrez Martín

Reviewer 2 Report

I feel like the manuscript has been greatly improved, however, I still have a few minor comments and suggestions.

Line 2: I would suggest changing the title (and any reference within the manuscript) to read "... from 2007 to 2020...". I suggest this because we are currently halfway through 2021 and there is no 2021 data within this manuscript. I also feel that it would be more fitting as this manuscript ages. When someone comes across this manuscript in 5, 10 years, having the date range explicitly defined would be more appropriate. 

Line 27: Again, I would change all "to date" references throughout to "to 2020"

Line 32: Insert "role" between "Lagomorphs" and "in" 

Line 33: Insert "is" between "it" and "also"

Line 50: Delete the space after tularensis and before the comma

Line 64: Should be spelled as "Sydney" and not "Sidney"

Line 134: Change "an" to "a"

Lines 136-137: Four-fold seroconversions are typically considered as positive and not "suspicious". What justification did you use for this designation?

Line 141: Move the hyphen to "real-time" for consistency

Line 143: Change title to read "Sample type and specific test(s) performed on human, domestic and wild animals, and water samples in this investigation"

Line 145: Why only spell out the abbreviation for MAT? If you define one, please define them all (MAT, PCR, ELISA)

Lines 148, 151-152, 171-173, 184-186, 191-192, 195-196: You use "inhabitants", "residents", and "denizens" interchangeably. I would pick one and use it throughout for consistency. If there is a specific reason why you use them at certain times, please specify. As I read through the manuscript, it just seemed random and confusing. 

Line 152: "Huge" is an inappropriate word here. "Significant"? If so, please show the statistics supporting, otherwise please use a different word. Perhaps "Notable" 

Line 157: I would delete "(both with 11%)" and just add "(11%) right after "crayfish" and "arthropods" 

Line 275: Delete the comma  after "strongly"

Author Response

REVIEWER 2 (second revisión)

Thank you for your great opinion about our second version of manuscript “Tularemia Outbreaks in Spain from 2007 to 2020 in Humans and Domestic and Wild Animals”.

As you can see above, we have changed “… to date” for “… to 2020” everywhere in the revised manuscript, such as you suggest. All your our considerations have been taken into account. Lastly, concerning your question from the lines 136-137, we can say that we used that criterion according to that previously established by the Laboratorio Central de Veterinaria, Algete (Madrid, Spain), dependent on the Spanish Ministry of Agriculture). In fact, we have added this information in lines 138-140, although this information already appereared in line 94 of the earlier revised article: “…according to the methods described by the Laboratorio Central de Veterinaria, Algete (Madrid, Spain) (dependent on the Spanish Ministry of Agriculture), such as isolation, identification, immunological, and molecular methods, based in turn on the World Health Organization [20]”.

Best regards.

César B. Gutiérrez Martín